# A Burn-In Apparatus for the ATLAS Tile Calorimeter Phase-II Upgrade Transformer-Coupled Buck Converters

Ryan Mckenzie [1,2,*] , Roger van Rensburg [1] , Seyedali Moeyedi [3], Edward Nkadimeng [1,2] , Stanislav Nemecek [4], Juan Buritica Yate [3], Haleh Hadavand [3] and Bruce Mellado [1,2]

[1]   School of Physics and Institute for Collider Particle Physics, University of the Witwatersrand, Private Bag 3, Johannesburg 2050, South Africa; bruce.mellado.garcia@cern.ch (B.M.)

[2]   iThemba LABS, National Research Foundation, P.O. Box 722, Somerset West 7129, South Africa

[3]   Department of Physics, University of Texas at Arlington, 502 Yates Street, Arlington, TX 76019, USA

[4]   Czech Academy of Sciences, Národní 3, 117 20 Staré Město, Czech Republic; stanislav.nemecek@cern.ch

[*]   Correspondence: ryan.peter.mckenzie@cern.ch

**Abstract:** The upgrade of the A Toroidal LHC ApparatuS (ATLAS) hadronic Tile Calorimeter (TileCal) Low-Voltage Power Supply (LVPS) forms a part of the Phase-II Upgrade preparations undertaken by the ATLAS experiment for the data taking during the High-Luminosity Large Hadron Collider era. This paper serves to provide a detailed overview of the development of a Burn-in test station for an upgraded LVPS component known as a Brick. The production, quality assurance testing, and all associated apparatus are being jointly undertaken by the University of the Witwatersrand (Wits) and the University of Texas at Arlington (UTA). These Bricks are radiation-hard transformer-coupled buck converters that function to step-down bulk 200 VDC power to the 10 VDC required by the on-detector electronics. To ensure the high reliability of the Bricks, once installed within the TileCal, a Burn-in test station has been designed and built. The Burn-in station functions to implement a Burn-in procedure on eight Bricks simultaneously. This procedure subjects the Bricks to sub-optimal operating conditions, which function to accelerate their ageing, as well as to stimulate failure mechanisms. This results in elements of the Brick that would fail prematurely within the TileCal failing within the Burn-in station or experience performance degradation that can be detected by follow-up testing effectively screening out the non-performative sub-population. The Burn-in station is of fully custom design in both its hardware and software. The development of the test station will be explored in detail; the preliminary Burn-in procedure to be employed will be provided; the preliminary and final commissioning of the test station will be presented. The paper will culminate in the presentation and discussion of the Burn-in of a V8.4.2 Brick and the future outlook of the project.

**Keywords:** ATLAS; TileCal Phase-II Upgrade; detector low-voltage power supply; transformer-coupled buck converters; quality assurance testing; Burn-in testing





## 1. Introduction

In the year 2029, the start of the operation of the High-Luminosity Large Hadron Collider (HL-LHC), located at the European Organization for Nuclear Research (CERN), is planned with a foreseen peak luminosity of $5 \times 10^{34}\,\mathrm{cm}^{-2}\,\mathrm{s}^{-1}$ in order to collect $4000\,\mathrm{fb}^{-1}$ of proton–proton collisions by the end of its operation [1]. The resulting HL-LHC environment has necessitated the Phase-II Upgrade of the A Toroidal LHC ApparatuS (ATLAS) general-purpose detector [2,3]. The ATLAS Hadronic Tile Calorimeter (TileCal) sub-detector will similarly receive an upgrade to both its on- and off-detector electronics to meet the requirements of a 1 MHz trigger and higher ambient radiation and to ensure better performance under high pile-up conditions [4,5].

The TileCal is a sampling calorimeter that forms the central region of the hadronic calorimeter of ATLAS and is located at $|\eta| < 1.7$ and $2.28\,\mathrm{m} < r < 4.23\,\mathrm{m}$. The ATLAS

experiment makes use of a right-handed coordinate system with its origin located at the nominal Interaction Point (IP) in the centre of the detector and the z-axis along the beam pipe. The x-axis points from the IP to the centre of the LHC ring, and the y-axis points upwards. Cylindrical coordinates (r,$\phi$) are used in the transverse plane, where $\phi$ is defined as the azimuthal angle around the z-axis. The pseudorapidity is defined in terms of the polar angle $\theta$ as $\eta = -ln \tan(\theta/2)$. The detector performs several critical functions within the ATLAS experiment such as the measurement and reconstruction of hadrons, jets, hadronic decays of $\tau$-leptons, and missing transverse energy [6]. It also contributes to muon identification and provides inputs to the Level-1 calorimeter trigger system [7,8]. The readiness of the TileCal for Run1 of the LHC, which took place during 2011 and 2012, was presented in a paper by the ATLAS collaboration [9].

The TileCal is physically partitioned into three cylindrical barrel regions along the beam axis, as seen in Figure 1. These are known as the Long Barrel located at $|\eta| < 1.0$ and Extended Barrels, of which there are two, located at $0.8 < |\eta| < 1.7$. Each barrel region consists of 64 wedge-shaped modules that cover the azimuthal angle $\triangle\phi \sim 0.1$ rad. The modules are composed of plastic scintillator tiles, functioning as the active media, inter-spaced by steel absorber plates. The ultraviolet light produced by the particle–matter interaction is transported by wavelength shifting fibres to Photo-Multiplier Tubes (PMTs), thereby producing signals that are sent to the on-detector Front-End (FE) electronics. The PMTs and FE electronics of a module will be housed within a new configuration. This configuration makes use of three-meter-long drawers, known as Super-Drawers (SDs), which are located at the outer radius of the TileCal. The Long and Extended Barrel modules utilise two SDs and one SD, respectively, resulting in a total of 256 SDs.

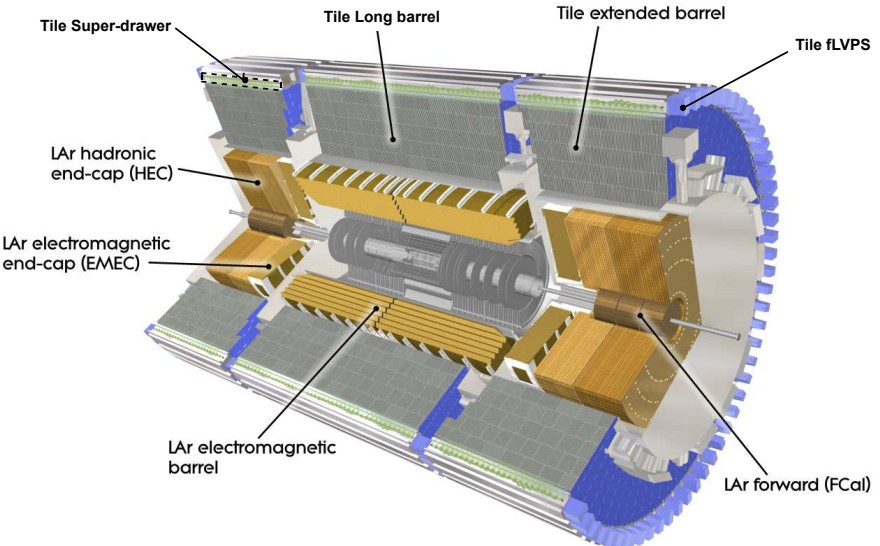

**Figure 1.** Computer-generated image of the ATLAS experiments inner-barrel [10]. The 256-finger Low-Voltage Power Supplies (fLVPSs) are located around the outer circumference of the TileCal, each of which provides low-voltage power to the front-end readout electronics located within a TileCal Super-Drawer.

A Low-Voltage Power Supply (LVPS) is located adjacent to the SD that it services and is housed within a shielded container known as a finger. Each finger LVPS (fLVPS) contains a fuse board, an Embedded Local Monitoring Board (ELMB) and its associated motherboard, a wiring harness, as well as eight transformer-coupled buck converters known as Bricks [11]. Each of the eight Bricks function to step-down the 200 V DC power received from off-detector power supplies to the 10 V DC power required by the Phase-II Upgrade FE electronics.

## 2. fLVPS Brick History, Design, and Function

All of the TileCal fLVPS Brick versions are of an iterative design. The V6.5.4 Brick was the first to be installed and operated within the TileCal in 2007 [12]. The iterative nature of the fLVPS Brick design is motivated not only due to the success of previous designs and the inherent benefits of refining a well-understood system, but also due to the stringent design limitations imposed by their unique operating environment and location. One such limitation is that the overall physical dimensions of the fLVPS are set at 175 mm × 170 mm × 156 mm by the size of the available internal volume of the "fingers", where they must fit. This results in a strong limitation being placed on the form factor of the Bricks. This restriction is still imposed for the Phase-II Upgrade Bricks as the volume of the fingers will not be altered during the upgrade.

The V6.5.4 Bricks functioned well at first, but began to exhibit a sensitivity for trips, which became apparent as the luminosity of the LHC was sequentially increased during commissioning. A trip is defined as a "spontaneous" event in which a Brick switches off. This trip rate scaled with the luminosity of the beam, further indicating that the performance of theV6.5.4 Brick would only further deteriorate and was, therefore, not fit for the purpose [13]. To address the issue of trips, as well as implement additional design changes motivated by experience gained with the V6.5.4 Bricks, the V7.5.0 Bricks were designed and subsequently installed within the TileCal in 2013. The V7.5.0 Bricks are to remain within the TileCal up until the Phase-II Upgrade, which will take place during the Long-Shutdown 3 (LS3).

During LS3, the latest V8.6.0 Bricks will be installed (pending final review). The replacement of the V7.5.0 Bricks is motivated by the high-luminosity environment that will be created by the HL-LHC, which necessitates more-stringent radiation hardness requirements for the active components. The TileCal detector will also take the opportunity to implement further design improvements of the low-voltage power distribution system. These improvements are the introduction of a third stage to the system and the introduction of tristate functionality.

The third stage of the power distribution system takes the form of point-of-load voltage regulators, which undertake the final step-down of the 10 VDC power received from the Bricks to the voltages required by the local front-end electronic circuitry. The introduction of this stage allows for a single Brick variant with a homogeneous output voltage as opposed to the V6.5.4 Brick, which had eight sub-variants, each of which had an output voltage tailored to the specific on-detector electronics that it powered. This resulted in not only increased production costs, but also made the storage of spares more cumbersome.

The newly introduced tristate functionality refers to the capability of the LV power distribution system control the on/off state of an individual Brick utilising a tristate voltage signal, which is routed to the Bricks' startup circuitry. This is a significant improvement as the on/off control of the legacy LV system was at the granularity of half of an fLVPS. One such implication of the legacy control was that the front-end electronics of a half a super-drawer would need to be power cycled even if only a portion of the drawer was exhibiting errors associated with single-event upsets. Although no negative behaviour was "observed" due to the needless power cycling of the font-end electronics, it is nonetheless undesirable.

As shown in Figure 2, the Phase-II Upgrade Brick, of which there will be a total of 2048 installed within the TileCal, provides a nominal output current of 2.3 A at 10 VDC.

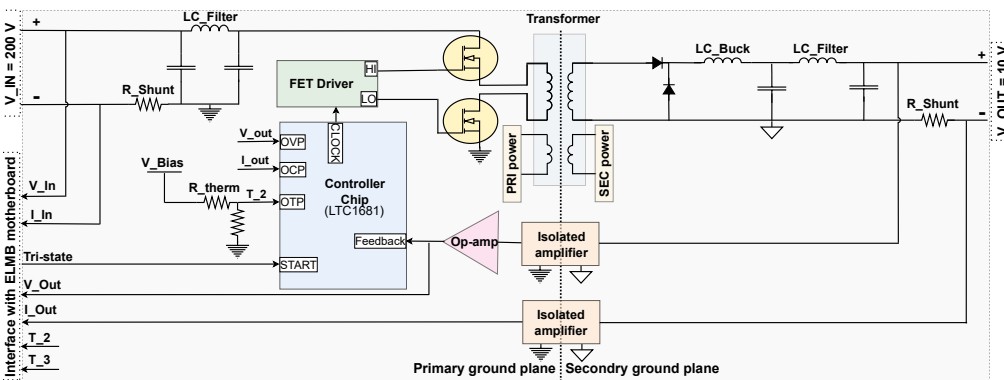

**Figure 2.** Functional block diagram of an fLVPS Brick.

The Bricks' design is centred on the Analog Devices LT1681 controller chip, which is a pulse-width modulator that operates at a fundamental frequency of 300 kHz. The pulse-width modulation is controlled by two inputs. The first input is received from a slow feedback path that monitors the output voltage with a bandwidth of approximately 1 kHz. This feedback path makes use of an isolation amplifier, ensuring galvanic isolation of the primary and secondary sides. The second input is a fast feedback path that monitors the current through the low-side transistor on the primary side of the Brick. The LT1681 provides an output clock to a Field Effect Transistor (FET) driver, which performs the switching of the two MOSFETs located on the primary side.

The design employs synchronous switching of both MOSFETs, that is both the high-side and low-side MOSFETs turn on and conduct for the duration that the output clock is in the high state, and both are off when the clock is in the low state. When the FETs conduct, the input current at 200 V DC flows through the primary windings of the transformer, which, in turn, transfer energy to the secondary windings. The primary to secondary winding ratio is 14:1, and the transformer has a nominal power loss of 3.95 W. The transformer serves additional purposes in that it provides galvanic isolation of the primary and secondary sides of the Brick and also has additional windings for other auxiliary power supplies on the Brick, required for control and monitoring purposes.

A buck converter is implemented on the secondary side of the Brick. The secondary (output) side also contains an inductor–capacitor stage for the filtering of high-frequency noise, which would otherwise be propagated to the front-end electronics. The V8.6.0 Brick utilises the same inbuilt protection circuitry implemented on previous iterations of the Brick, albeit with different trip points. That is, the design utilises three types of inbuilt protection circuitry, Over-Voltage Protection (OVP), Over-Current Protection (OCP), and Over-Temperature Protection (OTP). These circuits, if activated, initiate a shutdown of the Brick. Their activation depends on preset thresholds, which are discussed in Section 5.

The temperature measurements are taken by two thermistors located at the different positions that are illustrated in Figure 3. Thermistor T2 is adjacent to the primary-side switching MOSFETs of the Brick, whereas Thermistor T3 is located adjacent to the LC buck on the secondary side. The thermistors are located on the underside of the Brick.

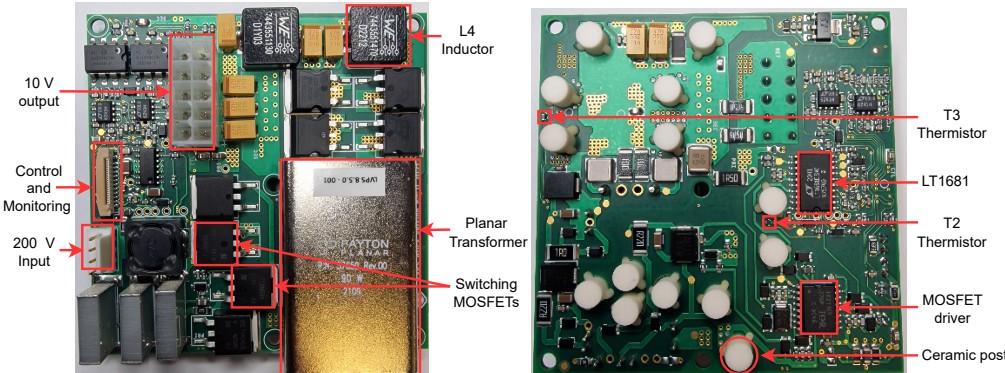

**Figure 3.** Top view (**left**) and bottom view (**right**) of a V8.5.0 Brick produced at Wits. The only difference between all Bricks produced by Wits and the UTA is the use of aluminium-oxide (appearing as white) or aluminium-nitride (appearing as grey) ceramic posts, respectively. The replacement of the switching MOSFETs to a more-efficient variant (reduced switching losses) and the replacement of the L4 inductor to a component with a form factor more conducive to the effective thermal coupling to the ceramic post below constituted the major upgrade of the V8.4.2 to the V8.5.0 Bricks in relation to efficiency and thermal performance. The LT1681 and MOSFET driver have a gap pad placed between them and a cooling plate during the assembly of the fLVPS to facilitate improved cooling.

## 3. Quality Assurance Testing of the Phase-II Upgrade fLVPS Brick

Access to the LVPS Bricks is of the order of once per year as the ATLAS detector and is required to be in the open position during the Year-End Technical Stop (YETS). Therefore, any Bricks that experience a permanent failure will remain in the same state until the upcoming YETS. This results in one-eighth (sixth) of a module's Front-End (FE) electronics, in the case of a single Brick failure, being offline for a commensurate period of time for a long (extended) barrel module, respectively. The off-line FE electronics are entirely unable to collect collision data. Due to this, the reliability of the Bricks is of the utmost importance. A failure is defined as the permanent inability of the fLVPS to provide power to its associated front-end electronics, necessitating replacement.

The reliability of an electronic device, such as a Brick, can differ from its predicted design reliability due to latent and patent defects associated with its production and that of its components. Quality assurance testing is to be undertaken on all fLVPS Bricks post-manufacturing to address this phenomenon [14], the purpose of which is to increase the reliability of the surviving population by screening out the sub-population of Bricks that exhibit such defects. As seen in Figure 4, the quality assurance procedure is composed of five distinct tests, namely an automated visual inspection, X-ray scan, initial testing, Burn-in, and final testing. Each test is tailored to ensure the high reliability of the Brick population, once installed on the detector, by detecting and/or stimulating any patent or latent defects that are present.

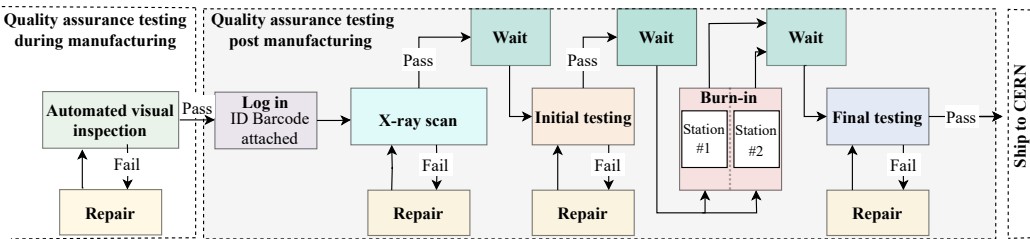

**Figure 4.** Block diagram illustrating the finalised fLVPS Brick quality assurance procedure.

Upon completion, every Brick receives a unique identification barcode in order to allow for the ease of tracking during its lifetime in the experiment. The Bricks then undergo an automated visual inspection, whereby visible manufacturing defects are identified and corrected. The Bricks proceed to the next step, whereby they receive an X-ray scan, which

focuses on the identification of possible solder outgas voiding that may have occurred between certain components during the reflow process, such as the primary-side switching FETs and the ceramic post to which they are affixed. The motivation for the detection of these voids is due to their propensity to reduced thermal conductivity between the electronic component tab and the ceramic post at the thermal interface.

The initial and final testing are identical. Each utilise the same test apparatus and procedure and are so named due to their occurrence before or after Burn-in, respectively [15]. Both tests measure and evaluate various Brick performance metrics and function to ensure that the Brick under test is operating within its design specifications. Undertaking performance testing both before and after Burn-in provides valuable insight into any performance degradation resulting from the Burn-in procedure while also serving to screen out Bricks that have been permanently damaged. However, Bricks that have latent defects that function marginally may not be detected by performance testing alone and would be installed within the TileCal. These Bricks would fail early in the life of the Phase-II Upgrade TileCal and give rise to a high initial failure rate, commonly observed in the electronic infant mortality region.

System-level Burn-in of all manufactured Bricks is to be undertaken. That is, the Bricks will undergo Burn-in once fully assembled and operational. Burn-in is required as some patent and/or latent defects may not have been detected by the initial testing. Burn-in is primarily focused on detecting patent defects that appear during the early life of the Bricks, but it should be noted that latent defects, which usually appear during normal operation, can be converted into patent defects via the application of external overstress [16]. Burn-in testing entails subjecting the Bricks to a Burn-in procedure in which they are exposed to overstress conditions, such as an increased operating temperature and applied load, in relation to their nominal operating conditions. This functions to stimulate failure mechanisms within the non-performant Brick population. These Bricks need not experience a catastrophic failure during Burn-in, but should be identifiable due to the identification of possible deterioration in their performance during final testing. It is worth emphasising that Burn-in does not improve the reliability of an individual Brick, but rather, the reliability of the surviving brick population as any identifiable non-performant Bricks are removed from the population.

Efforts will be made to repair the Bricks removed from the population. The repair of the Bricks is motivated by the fact that they have a high per-unit cost and, more importantly, is due to the fact that the active components used in their production are essentially one of a kind, as they are independently certified for radiation hardness in production lots. These lots, containing a fixed unit number, are approved for use within the production. The Bricks will be subjected to the quality assurance procedure again if repaired. This process will repeat until the Brick passes the entire quality assurance procedure or is deemed irreparable. The Burn-in apparatus employed is a custom design due to the custom nature of the Bricks that it is tasked with testing. This testing procedure is referred to as the Burn-in procedure in the subsequent paragraphs. The development of the Burn-in station is considered below.

## 4. Custom Phase-II Upgrade fLVPS Brick Burn-In Station

The Burn-in station, pictured in Figure 5, is an iterative design similar to that of the Bricks and is tasked with applying a Burn-in procedure. The key design requirement for the latest Burn-in station is that it is able to consistently apply the Burn-in procedure to each of the Phase-II Upgrade Bricks throughout the testing of the entire Brick population. Due to the scale of the Brick quality assurance testing, further requirements are imposed as many junior researchers will be involved in the undertaking. That is, it is a necessity that the operation of a Burn-in station be both simple and safe, allowing its use by non-experts.

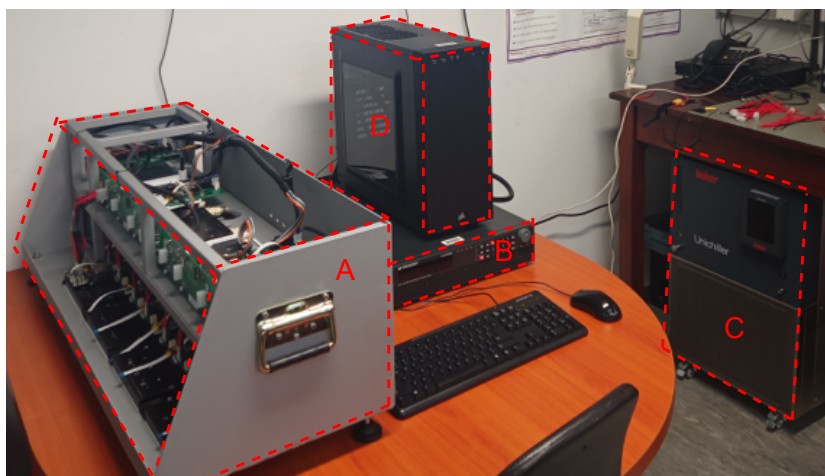

**Figure 5.** Labelled image of a Wits Burn-in test station. The test-bed (A), with its perspex cover and lid removed, can be seen on the left-hand side of the image. Input power to the Bricks is provided by the programmable DC power supply (B). The control of the applied Brick Burn-in temperature parameter is undertaken by a water chiller, which is also responsible for removing heat from the dummy loads (C). The control and monitoring of the Burn-in procedure and DC power supply are undertaken by the Burn-in LabVIEW application running on a personal computer (D).

The Burn-in station is composed of four distinct elements, namely the test-bed, cooling system, hardware, and software. The test-bed contains the majority of the Burn-in station hardware and the Bricks undergoing the Burn-in procedure, as seen in Figure A1. The test-bed is fully enclosed in a metal (main chassis) and perspex (top and front lid) box, reducing the impact of the outside environment on the operating temperature of the Bricks. The test-bed also has a safety role in that it prevents an unwitting user from coming into contact with any electrified elements. It is still foreseen that a safety interlock will be included on the front lid, which will shut-off the High-Voltage (HV) input to the Bricks should the lid be opened during the Burn-in procedure. The test-bed makes use of thermal and electrical insulation to prevent internal water condensation, as well as potential electrical shorts, while also being physically grounded.

An active cooling system is employed to control the operating temperature of the Bricks, a key Burn-in parameter, during Burn-in, as well as to remove heat produced by the dummy loads to which they provide power. The cooling system makes use of a commercial external water chiller, which brings distilled water to the desired setpoint before pumping it to the four Brick Cooling Plates (CPs) and, then, subsequently, to the four dummy load CPs, which are in a single cooling-loop. The implementation of a single cooling-loop allows for a single external water chiller to be used to control the Burn-in temperature, as well as cool the dummy load boards. The water chiller also has a total cooling capacity that allows it to control the temperature of two Burn-in test-beds. The benefit of this design is a reduction in the cost of the Burn-in stations, but this also results in additional complications with the cooling of the dummy load boards. The coolant temperature can be increased or decreased, thereby inducing a commensurate change in the temperature of the Bricks, allowing for the Brick Burn-in operating temperature to be set.

The Burn-in station hardware and software are discussed in detail in Appendices A and B, with the most important Burn-in performance measurements depicted in Table 1. The lack of a Temperature 1 measurement is an artifact of the removal of this measurement, which was used in previous iterations of the Brick. The measurements are taken by thermistors located at the 2nd and 3rd positions. Position 2 is adjacent to the primary-side switching MOSFETs of the Brick, whereas Position 3 is located adjacent to the LC buck on the secondary side.

**Table 1.** Burn-in station performance measurements and their origin. The measurements labels as used in the Burn-in LabVIEW Application (BLA) and shown in the Graphical User Interface (GUI) in Figure A7. The GUI or LabVIEW Front Panel shows the performance measurements of each Brick during a Burn-in procedure.

| Origin | Measurement | Label on BLA |
|---|---|---|
| Brick | Input voltage | Vin [V] |
| | Output voltage | Vout [V] |
| | Input current | Iin [A] |
| | Output current | Iout [A] |
| | Temperature 2 | T2 [°C] |
| | Temperature 3 | T3 [°C] |
| Dummy Load Board | Brick voltage at load | Vload [V] |
| | Brick current at load | Iload [A] |
| High-Voltage Supply | Input voltage | Vin [V] |
| | Input current | Iin [A] |

## 4.1. Hardware

The Burn-in station hardware is composed of a Personal Computer (PC), a 200 VDC High-Voltage (HV) power supply, custom-designed Printed Circuit Boards (PCBs), electrical wiring, cooling plates, and a test-bed. The PCBs are subdivided into four types, namely the Main Board (MB), the Brick Interface Board (BIB), the Load Interface Board (LIB), and the Dummy Load Board (DLB). The location of the various PCB types within the test-bed is illustrated in Figure 6.

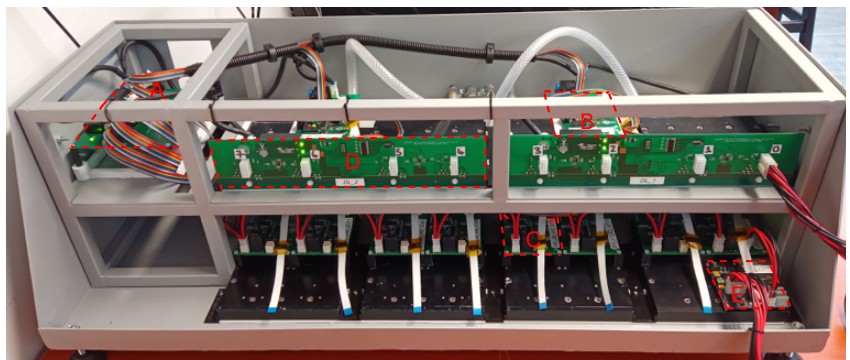

**Figure 6.** Labelled image of a Wits Burn-in station test-bed with its perspex covers removed. A single Brick undergoing Burn-in in Channel 0 can be observed (E). The main board (A) is connected to two Load Interface Boards (B), each of which is connected to an associated Dummy Load Board (D). The Main Board is also connected to eight Brick Interface Boards (C). The black cooling plates to which both a Brick and the Dummy Load Boards are affixed can also be observed. Note that only one of each PCB type is labelled.

The PCBs are supplied by 220 VAC mains power, which is rectified using DC–DC converters to supply regulated power to the active components. A simplified block diagram of the Burn-in station hardware is provided Figure A1. Based on the legacy designs developed by Argonne National Laboratories (ANL), the Burn-in station PCBs were redesigned by the University of Texas Arlington (UTA), with refinement and testing being undertaken by the University of the Witwatersrand (Wits).

*4.2. Software*

The Burn-in LabVIEW Application (BLA) and Programmable Integrated Circuit (PIC) firmware were originally developed by Argonne National Laboratory (ANL) in 2006 for the V6 Bricks [12,17]. Based on the legacy designs developed by ANL, the Burn-in station PCBs have been redesigned by the University of Texas Arlington (UTA) and developed by the University of the Witwatersrand (Wits). During the testing and development of the Burn-in station, the BLA underwent major revisions to improve code readability, while only minor changes were made to the HV power supply instrumentation drivers and firmware (ADC channel configurations) of the interface boards. The software required for the operation of a Burn-in station can be divided into three categories, namely the BLA, the PIC firmware, and the high-voltage programmable power supply instrumentation drivers.

## 5. Phase-II Upgrade fLVPS Brick Preliminary Burn-In Procedure

The Burn-in procedure subjects the Bricks to sub-optimal operating conditions, which function to stimulate failure mechanisms within the Bricks. This procedure is the subject of ongoing research. These conditions needs to fall within the extrema allowed for by the Brick design and operation. There are two reasons for this. The first is related to effective Burn-in, and the second is related to the implementation of the Burn-in procedure. Firstly, the failure mechanisms stimulated must be of the kind that can be experienced during operation within the TileCal such as an increased operating temperature. Secondly, the Burn-in parameters must be restricted to the parameters provided in Table 2 due to the Bricks' inbuilt protection circuitry. A Brick will initiate a trip (immediate shutdown) if the OCP or OTP trip points are reached during Burn-in. If the operating parameters are set to within the variance of the trip points, intermittent Brick trips would occur during the Burn-in procedure, which would not only be operationally taxing, but also bring the validity of the Burn-in results into question, whereas if the parameters are set higher than this variance, the Bricks will start and immediately trip so that Burn-in could not take place.

**Table 2.** Preliminary Phase-II Upgrade Brick Burn-in parameters, nominal V8.6.0 Brick operating parameters (over-voltage, over-current, over-temperature), and protection circuitry trip points.

| Parameter | Burn-In | Nominal | Protection |
|---|---|---|---|
| Operating temperature | 60 °C | 35 °C | 70 °C (OTP) |
| Applied load | 5.0 A | 2.3 A | 6.9 A (OCP) |
| Output voltage | 10 V | 10 V | 11.85 V (OVP) |
| Run-time | 8 h | - | - |

The run-time of the Burn-in procedure is an important parameter. One needs to weigh the efficacy of the Burn-in process against the practicality of undertaking the Burn-in of a large population of Bricks within a finite time period. The above points combined with previous experience obtained through the Burn-in of the V7.5.0 Bricks were considered, resulting in the Burn-in parameters provided in Table 2 being selected. The 8 h run-time originated from the development of the V6.5.4 Brick and its associated quality assurance procedure by Argonne National Laboratory. The explicit calculations in which this run-time was determined were unfortunately not recorded, with information provided only in the form of internal presentations of the Burn-in procedure. As this Burn-in duration has been successful when applied to two previous Brick versions, it was considered as a good starting point for the preliminary Burn-in procedure. That being said, numerous components have been changed in the subsequent Brick versions, and a naive assumption of the equivalency of their behaviour during Burn-in presents a major risk. Therefore, a quantitative analysis of the Burn-in duration and its efficacy are the subject of ongoing research.

A generalised Phase-II Brick Burn-in procedure thermal profile is illustrated in Figure 7. The $T_2$ temperature value originates from a thermistor located in between the primary-side MOSFETs. This thermistor, which records one of two on-Brick temperatures, was

selected as it is utilised in the Brick OTP circuitry in which it measures the highest on-Brick temperatures originating from the MOSFET switching losses. The thermal profile can be sub-divided into three distinct regions. These are known as the rise, soak, and fall regions. At the onset of the Burn-in procedure, the Bricks are switched on and put under a 5 A load. The Bricks and the cooling system are in thermal equilibrium at 20 °C at this point. This lower bound was selected due to it being the average on-detector temperature of the Bricks when they are not in operation. The temperature steadily climbs via the action of the cooling system, increasing the temperature of the heat-sinks. This process is assisted by the Bricks' self-heating due to inherent power conversion inefficiency. The characteristic curve is a result of the heat capacity of the system and the conductive, as well as radiative losses. The soak region begins after time $T_{rise}$ and continues for a total time period known as the soak time at the Burn-in temperature of 60 °C. The Burn-in procedure run-time is taken as the sum of the rise and soak time. This is because the Brick must remain under load during the entirety of the Burn-in procedure. At the eight-hour mark, the Burn-in procedure is terminated with the Bricks being switched off and brought down to their starting temperature. The time that it takes for the Bricks to be brought down to their starting temperature is known as the fall time.

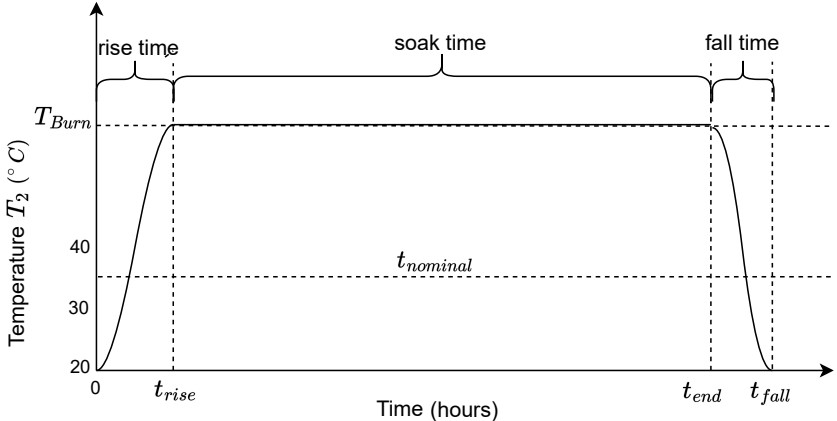

**Figure 7.** Generalised Phase-II Brick preliminary Burn-in procedure thermal profile.

## 6. Commissioning of the fLVPS Brick Burn-In Station

The commissioning of the Burn-in station is partitioned into two distinct commissioning procedures. Preliminary commissioning is undertaken to ensure that the test station is safe for operation and that it is able to apply a Burn-in procedure. Final commissioning focuses on the calibration of the test station.

### 6.1. Preliminary Commissioning

Preliminary commissioning begins with a visual inspection of all safety-critical hardware. Particular emphasis is placed on the temperature control system and all electrical connections due to the inherent danger posed by their close proximity. This process consists of a review of all coolant pipe joints, coolant levels within the water chiller, electrical and thermal insulation of the cooling plates, grounding of the test-bed, inspection of all cable connections and the PCBs to which they are connected, and ensuring that the mica films located between the dummy load power MOSFET tabs are correctly affixed. It was found that the thermostats located above the dummy load MOSFETs, which act as inbuilt thermal protection circuitry, have a propensity to trip during normal operation. As a result, the thermostats were replaced with an identical package with a trip point of 130 °C, as opposed to the previous thermostats, which had a trip point of 110 °C. The maximum allowable MOSFET junction temperature was taken into account when making this design change.

Thermography was used as a part of the commissioning process due to its ability to provide a macroscopic view of the thermal behaviour of the Burn-in station. The findings of the thermal review were corroborated by the use of an external thermocouple. Key

areas such as the dummy load power MOSFET and the primary-side switching MOSFETs of the LVPS Bricks were reviewed with their operating temperatures being measured at 30.2 °C and 62.12 °C, respectively. The cooling plates were found to be at 38 °C (Figure 8, bottom-right) which was 2.0 °C lower than the temperature setpoint of the water chiller. This deviation will be taken into account when applying the Burn-in procedure.

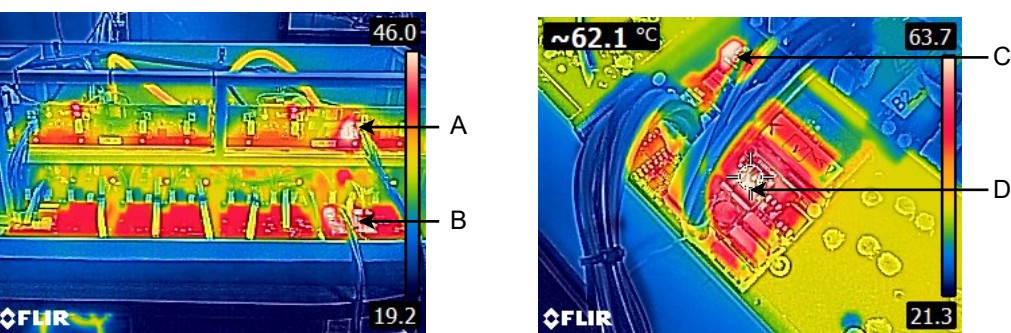

**Figure 8.** Thermal image of a Burn-in station, with its covers removed, during the Burn-in of a Brick located in Channel 1 of 8 (**left**). A dummy load voltage-controlled current sink (A) can be seen converting the power received from a V8.4.2 Brick undergoing Burn-in (B) into heat. A thermal image of a special sub-variant of a V8.4.2 Brick undergoing Burn-in (**right**). The target is centered on heat generated by the primary-side MOSFETs (D). This sub-variant made use of new, more efficient, switching MOSFETs, which were adopted in all subsequent versions. A second hot spot can be observed on the top corner of the Brick (C). This is an inductor that forms part of the buck converter on the secondary side. This component was changed to one with a form factor that allowed for better thermal coupling to the ceramic post below it. This new (L4) inductor can be seen in Figure 3.

### 6.2. Final Commissioning

The Burn-in station records and displays numerous behavioural parameters. The purpose of these measurements is to ensure that the Burn-in procedure is being correctly applied while also providing insight into the performance of the Bricks during Burn-in. The raw data originate from two distinct sources, as can be seen in Table 1. That is, the performance data are measured by the Bricks themselves and the dummy load boards to which they provide power. The data produced by these two sources take two types of paths to the PC, which are illustrated in Figure A3. The first path type considered consists of raw analog data measured by a Brick, or dummy load board, which is then converted by the ADC of the associated interface board into a representative serial bit stream. The bit stream is then transmitted to the BLA, where it is scaled, displayed, and logged. The scaling of a Brick's raw digital data is illustrated in Figure 9. It was assumed that the relationship between the raw analog data and the corresponding digital data is linear within a small deviation from nominal. Therefore, a linear function $f(x_j)$ was used to perform the scaling of the *j-th* raw data $x_j$. The scaling of $x_j$ was achieved by the selection of appropriate calibration constants $a_j$ and $b_j$, which correspond to a gain and offset, respectively. The selection of appropriate gain and offset values to ensure equality between the physical values and the recorded and displayed values constitutes the calibration of the Burn-in station.

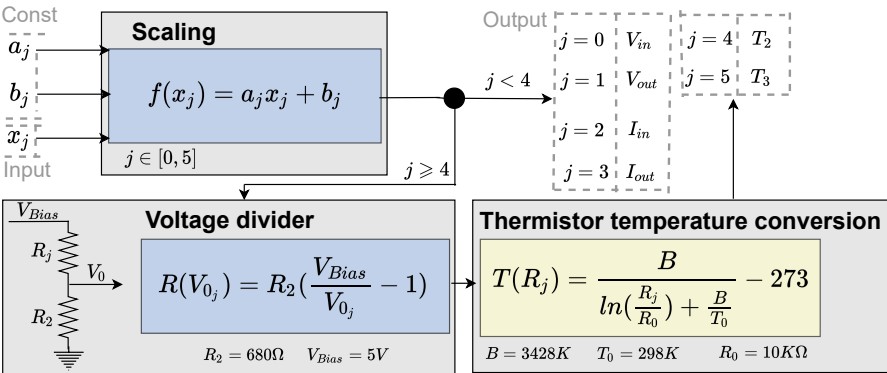

**Figure 9.** Block diagram illustrating the linear scaling of digitised Brick Burn-in parameters. The temperature parameters, $j \geq 4$, require additional processing in that the digitised thermistor voltage is converted into its equivalent resistance and, then subsequently, into its operating temperature.

The calibration constants are unique for each *j-th* value of a Brick and for each of the eight Bricks. This procedure is followed for $j < 4$. For the remaining two measurements, the data need to be converted from a voltage $(V_0)$ that is representative of the temperature of the $T_2$ or $T_3$ thermistors (Part No. NCP18XH103J03RB, Murata, Kyoto, Japan) into their actual temperature. This takes the form of two additional steps. The first step uses the scaled voltage to determine the thermistors' resistance $R_j$. This resistance is then used to calculate the thermistors' temperature. The relationship between the voltage received from the voltage divider and the thermistors' operating temperature is illustrated in Figure 10 left. The thermistor B value of 3428 k was chosen with the thermistor calibration curve for this value illustrated in Figure 10, right. The dummy load raw data undergo the same process of scaling and associated calibration as the current and voltage measurements of the Bricks for each of the eight dummy load channels (1 channel per Brick).

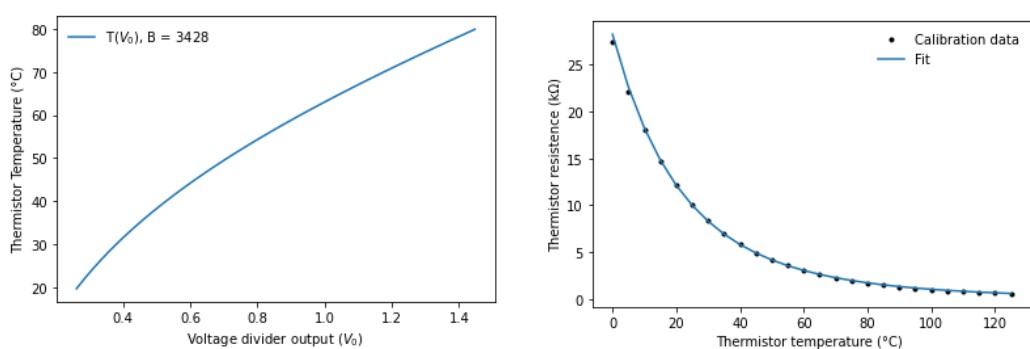

**Figure 10.** Thermistor temperature as a function of the voltage divider output voltage (**left**) and its calibration curve (**right**).

## 7. Summary and Outlook

A Burn-in apparatus, referred to as the Burn-in station in this paper, for the ATLAS Tile Calorimeter Phase-II Upgrade transformer-coupled buck converters has been developed and manufactured. The motivation for and application of Burn-in testing were provided within the context of the improved reliability of the fLVPS Bricks. The history and function of the LVPS Bricks were furnished with the emphasis being placed on their inbuilt protection circuitry, which sets limits on the possible Burn-in parameters. The four distinct elements of a Burn-in station, namely the test-bed, cooling system, hardware, and software, were introduced with detailed information provided in the Appendix. The Phase-II Brick Burn-in procedure was provided in which the Burn-in parameters were stated and motivated. The generalised Phase-II Brick Burn-in procedure thermal profile was also presented and

described with the paper, culminating in the explanation and presentation of the initial and final commissioning of the Burn-in station.

To further assess if the final commissioning of the Wits Burn-in station was successful, an entire Burn-in procedure was applied to a V8.4.2 Brick. A special V8.4.2 Brick, with the "new" higher-efficiency MOSFETs and L4 inductor as updated in the V8.5.0 Brick, was used. This was performed so that the V8.4.2 Brick provided an approximate analog to the V8.5.0 Brick in terms of thermal characteristics and efficiency. This was necessitated as the V8.5.0 Brick shown in Figure 3 (and five others like it) had not been produced at the time of testing. The low sample size (one Brick) of the V8.4.2 Brick was due to the majority of the South African Bricks being sent directly to CERN for use in the TileCal Phase-II Upgrade test beam campaigns. In order to increase the sample size and corroborate the current findings, the measurements presented below will be repeated with the six V8.5.0 Bricks.

The measurements of the Bricks' T2 and T3 temperature, as well as their efficiency during the Burn-in procedure, provide a probe into not only the performance of the Bricks, but also that of the Burn-in station. The T2 and T3 thermal profiles depicting the Burn-in of a single V8.4.2 Brick are illustrated in Figure 11. Three key observations can be made when comparing these thermal profiles to the idealised thermal profile illustrated in Figure 7. The first observation is that there is a difference of 10 °C between the T2 and T3 thermistor values, which was expected. This difference is due to their location, illustrated in Figure 3, being adjacent to the switching MOSFETs and LC-buck, respectively. These components output a different amount of heat during operation. The second observation that can be made is that the starting temperature is not 20 °C, as motivated previously in Section 5. This is a consequence of the way in which the Burn-in temperature control is currently being implemented. Due to the Bricks not being 100% efficient, "self-heating" produces a $\Delta T$ between the thermostat values and cooling plate of approximately 20 °C. Therefore, to achieve the Burn-in temperature of 60 °C, the cooling plates need to be maintained at 40 °C. The thermal profile can be brought back into the specifications by either starting the cooling station at the commencement of the Burn-in procedure or programming the cooling station to ramp its temperature. The former ad hoc option is to be implemented, but development is anticipated for the latter. Finally, the trailing edge of the thermal profile is not visible. This is due to the data acquisition being terminated at the end of the Burn-in procedure. Although the thermal behaviour is well understood and should subsequently decrease to 40 °C, this will be addressed.

The efficiency of the V8.4.2 Brick during Burn-in is illustrated in Figure 12 (left). The high efficiency observed at the start of the Burn-in procedure, which sharply decreases, can be attributed to the RC time constant associated with the input current measurement circuitry and is, therefore, not indicative of the Brick's actual efficiency at startup. The input current has an inverse relationship with the efficiency with its behaviour at Brick startup seen in Figure 12 (right). A linear increased Brick efficiency, which also appears to be attributed to the input current measurement, can be observed. This behaviour is currently under investigation as a "steady-state" efficiency of approximately 70–75% is expected. This investigation will take the form of an external measurement of the Bricks' input current, as well as the application of a longer-duration Burn-in procedure to investigate whether or not the current stabilises.

A University of the Witwatersrand Burn-in station has been developed, manufactured, and commissioned. The station is poised for use within the pre-production of approximately 100 units, in the fourth quarter of 2023, at the University of the Witwatersrand (Wits). This will serve as a preparatory step for the full-production and quality assurance testing of 1024 units in 2024 at this institution, which is well within the current Phase-II Upgrade schedule. The preproduction will also allow for the refinement of the Burn-in procedure as a substantially larger dataset will be produced. A second identical Burn-in station is currently being produced to reduce the anticipated bottleneck at the Burn-in stage of the quality assurance procedure at Wits. The same procedure is being mirrored in parallel at the University of Texas at Arlington (UTA), which will be producing the same amount

of Bricks and undertaking identical quality assurance testing, with identical test stations, locally. Studies into the Burn-in duration, the observed Brick efficiency increase over time, and the refinement of the Burn-in station data acquisition are the subject of current work.

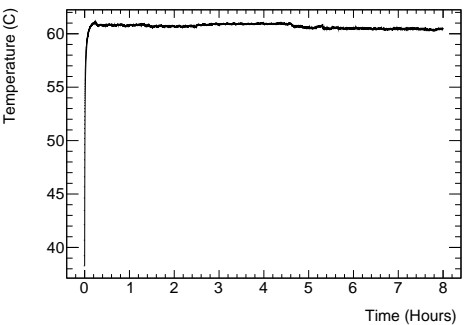
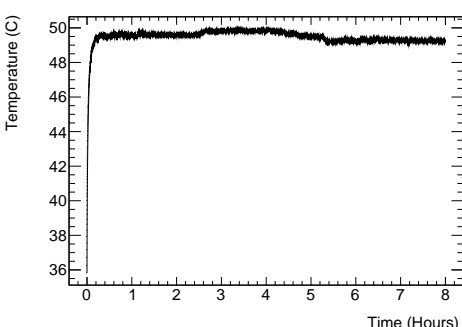

**Figure 11.** Thermal profile of a Phase-II Upgrade LVPS V8.4.2 Brick undergoing 8 h Burn-in testing. The T2 and T3 thermistor values are presented on the left and right, respectively. The lack of an anticipated trailing edge indicating the reduction of the Brick lower temperature bound is due to the cessation of data recording at the end of Burn-in procedure. This behaviour is being addressed in the latest Burn-in LabVIEW Application.

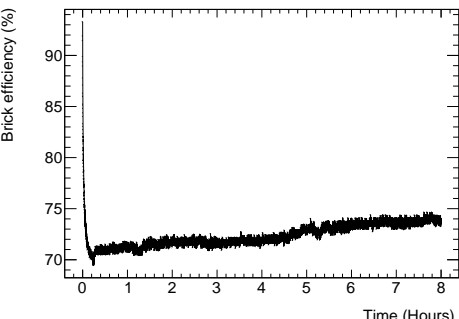
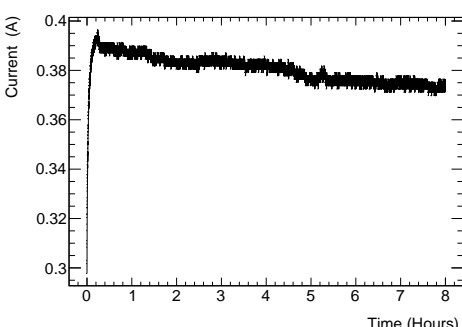

**Figure 12.** Efficiency plot of a Phase-II Upgrade LVPS V8.4.2 Brick undergoing Burn-in testing (**left**). The Brick's efficiency is defined as the ratio of the output and input power expressed as a percentage. The behaviour of the Brick's input current measurement over the Burn-in procedure (**right**).

**Author Contributions:** Conceptualisation, R.M., R.v.R. and S.M.; formal analysis, R.M.; funding acquisition, H.H. and B.M.; investigation, R.M., R.v.R., S.M. and J.B.Y.; methodology, R.M., R.v.R. and S.M.; project administration, H.H. and B.M.; resources, S.M., E.N. and S.N.; software, R.v.R. and S.M.; supervision, R.v.R., S.M. and S.N.; validation, R.M., R.v.R. and S.M.; visualisation, R.M. and R.v.R.; writing—original draft, R.M. and R.v.R.; writing—review and editing, R.M. and R.v.R. All authors have read and agreed to the published version of the manuscript.

**Funding:** This research was funded in part by the National Research Foundation of South Africa, Grant Number 123017.

**Data Availability Statement:** The data presented in this study are available on request from the corresponding author. The data are not publicly available as the TileCal Phase-II Upgrade database is still being commissioned.

**Acknowledgments:** The authors would like to acknowledge the South African National Research Foundation, the Department of Science and Innovation, and the SA-CERN program for their funding contribution without which this research could not have been conducted. The ATLAS Hadronic Tile Calorimeter also requires acknowledgement for providing their expertise and time within the context of the LVPS Brick project as a whole.

**Conflicts of Interest:** The authors declare no conflict of interest. The funders had no role in the design of the study; in the collection, analyses, or interpretation of the data; in the writing of the manuscript; nor in the decision to publish the results.

**Abbreviations**

The following abbreviations are used in this manuscript:

| | |
|---|---|
| ATLAS | A Toroidal LHC ApparatuS |
| TileCal | ATLAS hadronic Tile Calorimeter |
| HL-LHC | High-Luminosity Large Hadron Collider |
| CERN | European Organization for Nuclear Research |
| LVPS | Low-Voltage Power Supply |
| DC | Direct Current |
| PMT | Photo-Multiplier Tube |
| FE | Front-End |
| SD | Super-Drawer |
| FET | Field Effect Transformer |
| OVP | Over-Voltage Protection |
| OCP | Over-Current Protection |
| OTP | Over-Temperature Protection |
| MOSFET | Metal–Oxide Semiconductor Field-Effect Transistor |
| CP | Cooling Plate |
| PC | Personal Computer |
| PCB | Printed Circuit Board |
| MB | Main Board |
| IB | Interface Board |
| BIB | Brick Interface Board |
| LIB | Load Interface board |
| DLB | Dummy Load Board |
| VCCS | Voltage-Controlled Current Sink |
| BLA | Burn-in LabVIEW Application |
| USB | Universal Serial Bus |
| PIC | Programmable Integrated Circuit |
| UART | Universal Asynchronous Receiver/Transmitter |
| DTR | Data Terminal Ready |
| RTS | Request To Send |
| INH | Inhibit |
| ADC | Analog-to-Digital Converter |
| ELMB | Embedded Local Monitoring Board |
| SPI | Serial Peripheral Interface |
| SCK | Serial Clock |
| SDI | Serial Data Input |
| SDO | Serial Data Output |
| CS | Chip Select |
| DAC | Digital-to-Analog Converter |
| GND | Ground |
| ANL | Argonne National Laboratory |
| UTA | University of Texas at Arlington |
| Wits | University of the Witwatersrand |
| HV | High-Voltage |

## Appendix A. Custom fLVPS Brick Burn-In Station Hardware

*Appendix A.1. Introduction*

As illustrated in Figure A1, there is one MB per Burn-in station responsible for addressing and demultiplexing (one-to-many signal channelization) of the interface boards (BIBs and LIBs). The addressing and demultiplexing scheme allows serial communication from the Burn-in LabVIEW Application (BLA), running on a PC, to communicate via a

Universal Serial Bus (USB) to a universal asynchronous interface on the MB to the interface boards. A program running on the Programmable Integrated Circuit (PIC) of the MB polls each of the interface boards sequentially to perform a particular task.

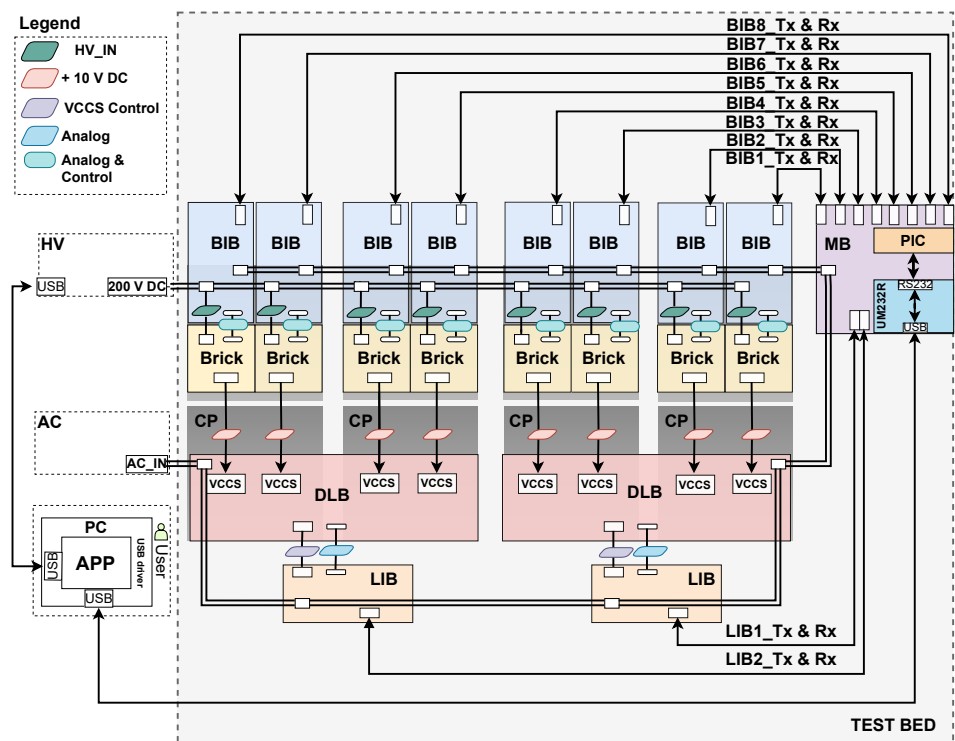

**Figure A1.** A simplified block diagram of the Burn-in station illustrating the test-bed and interconnections between various hardware components of the complete system.

There are eight BIBs per Burn-in station with one assigned to each of the Bricks undergoing the Burn-in procedure. The BIBs provide the control and monitoring functionality of their respective Bricks, while the LIBs provide control and monitoring of a Dummy Load Board (DLB), of which there are two per Burn-in station. A single Brick is connected to a BIB for the readout of Brick performance measurements, where the Brick output power is connected to a resistive load located on a DLB that incorporates closed-loop Voltage-Controlled Current Sink (VCCS) circuitry. The upcoming sections present a comprehensive analysis of the main hardware components and their functions providing an in-depth technical specification of the system.

*Appendix A.2. Main Board*

The Burn-in station comprises a single MB responsible for communicating to the BIBs and LIBs through an application-specific control and monitoring program developed in LabVIEW. As illustrated in Figure A2a, the BLA sends commands over a USB to RS232 converter (Part No. UM232R, FTDI, Jemstech (Pty) Ltd., Centurion, ZA, South Africa) for serial communication to the PIC. A Data Terminal Ready (DTR) signal line of the UM232R passes through an opto-coupler (Part No. 6N137S, Lite-on, Jemstech (Pty) Ltd., Centurion, ZA, South Africa) and is momentarily triggered once a communication link is established with the BLA, causing the PIC of the MB and interface boards to reset.

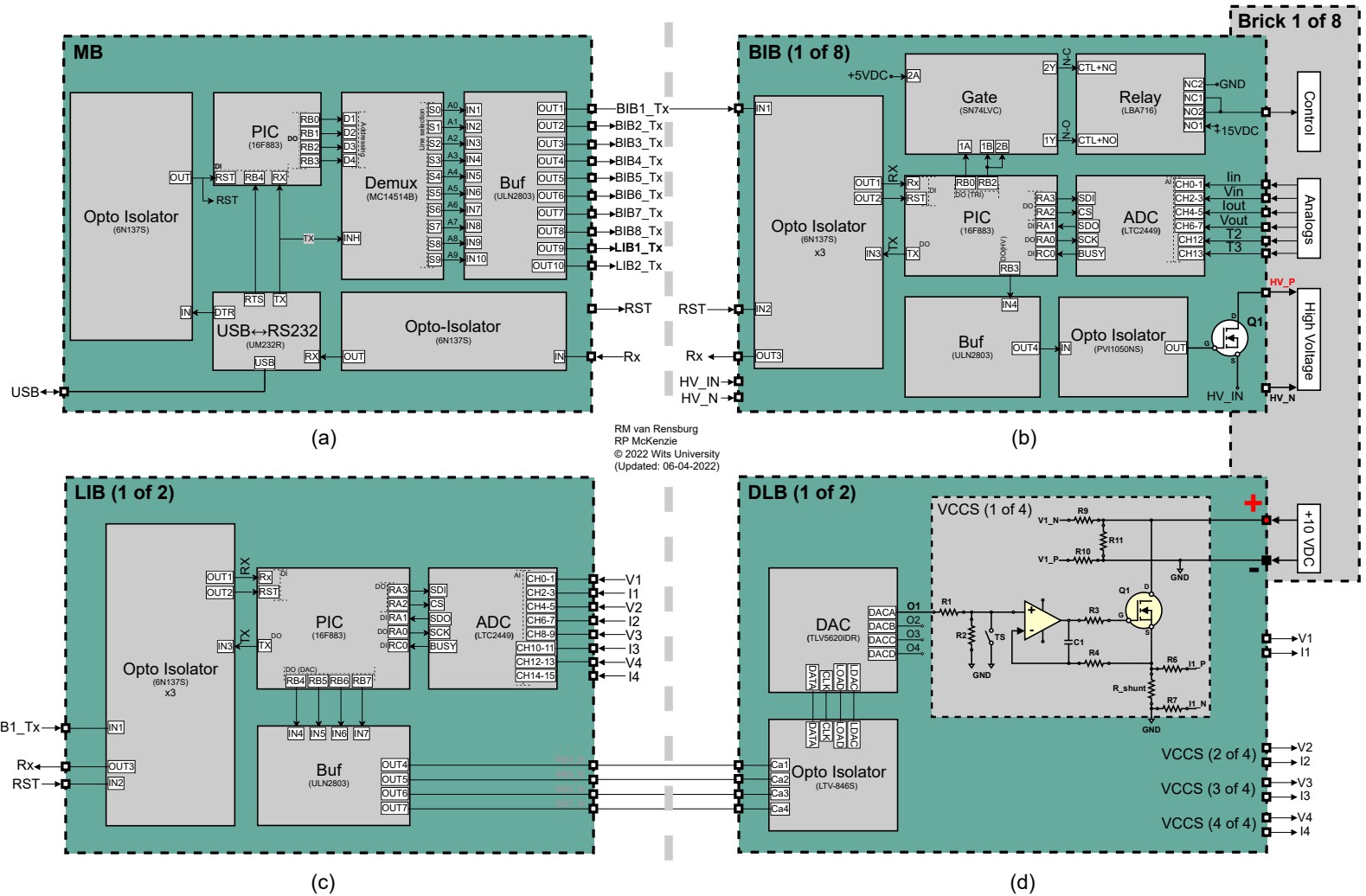

**Figure A2.** Simplified block diagram illustrating Burn-in station hardware components and interconnects. The (**a**) MB ensures bi-directional serial communication to the interface boards. A serial signal (BIB1_Tx...BIB8_Tx and LIB1_Tx...LIB2_Tx) is sequentially transmitted from the MB to their respective interface boards, where (**b**) represents one of eight BIBs and (**c**) one of two LIBs. There are two DLBs present per Burn-in station where (**d**) is one of two DLBs that controls four resistive loads. Note, four VCCSs are incorporated per DLB.

Universal Asynchronous Receiver Transmitter (UART) hardware flow control is partially introduced by asserting and deasserting a Request To Send (RTS) signal from the UM232R to address the interface boards using the select lines denoted by S0–S9 shown in Figure A2a. The assert and deassert of the RTS line is controlled by the BLA through the RTS digital output of the UM232R fed to a digital input RB4 of the PIC. Once the RTS line is asserted, the PIC firmware of the MB enters a conditional statement, which reads a serial signal (Rx) from the UM232R (Tx), which changes the states of the digital outputs RB0–RB3 that feed to the data lines D1–D4 of the demultiplexer (Part No. MC14514, Onsemi, Jemstech (Pty) Ltd., Centurion, ZA, South Africa). The data lines D1–D4 correspond to a four-bit address, which selects the line representing an interface board for communication over the desired select lines §0–S9 through the Inhibit line (INH) of the demultiplexer. While the RTS line is deasserted, addressing is complete and requests are sent by the BLA followed by USB to RS232 conversion to the selected interface board address A0–A9 to perform a particular task. Since the demultiplexer inverts the serial bit signal representing a character or string of characters representing a particular task for the interface boards to perform, buffer (Part No. ULN2803, Microelectronics, Jemstech (Pty) Ltd., Centurion, ZA, South Africa) circuits are implemented for buffering PIC control outputs and reverting the serial bit signal in its original state sent by the BLA. The BIBs and LIBs perform tasks such as enabling and disabling Bricks, setting load currents and reading analog channels for measuring Brick and load performance.

An opto-coupler (Part No. 6N137S, Lite-on, Jemstech (Pty) Ltd., Centurion, ZA, South Africa) with an open collector output acts as a line receiver and provides signal isolation of the UART signals transmitted from the interface boards. The UART signals from the interface boards are transmitted to the UM232R converter and ultimately the BLA over USB for monitoring and data logging.

*Appendix A.3. Interface Board*

A Burn-in station contains eight BIBs and two LIBs, which connect eight Bricks and two DLBs, respectively. A BIB provides the control and data acquisition of a Brick and a LIB provides the control and data acquisition of a dummy load. As illustrated in the block diagram of Figure A2b, each of the serial signals (BIB1_TX to BIB8_TX) transmitted from the MB have a dedicated BIB where an Analog-to-Digital Converter (ADC) reads directly from the Brick. Similarly, in Figure A2c, each of the serial signals (LIB1_TX to LIB2_TX) transmitted from the MB have a dedicated LIB, where the ADC reads from one of the two DLBs shown in Figure A2d. The control and data acquisition of the interface boards are explained in more detail in the upcoming sections.

Appendix A.3.1. Brick Interface Board Control

As discussed in Appendix A.2, a serial signal (from either one of the BIB1_TX...BIB8_Tx signals) transmitted by the BLA represents a particular task to perform by the PIC (Part No. 16F883, Microchip, Jemstech (Pty) Ltd., Centurion, ZA, South Africa). With reference to Figure A2b, the serial signal is first buffered through an opto-coupler (Part No. 6N137S, Lite-on, Jemstech (Pty) Ltd., Centurion, ZA, South Africa), which provides signal isolation from the MB and is received by an Rx input of the PIC. The Rx signal is then processed by the PIC firmware where a particular task is performed to read the analog measurements from a Brick.

To enable and disable a Brick, a logical AND gate (Part No. SN74LVC, Texas Instruments, Jemstech (Pty) Ltd., Centurion, ZA, South Africa) and a dual-solid-state relay (Part No. LBA716, IXYS, Jemstech (Pty) Ltd., Centurion, ZA, South Africa) are used to switch +15 VDC to a tristate input of a Brick. A tristate represents three voltage levels to start Bricks from an Embedded Local Monitor Board (ELMB), however, only an external 0 VDC and +15 VDC signal is needed to enable or disable a Brick, respectively, during a Burn-in procedure.

The control of a Brick may be dissected using a logic truth table as follows: When both digital outputs RB0 and RB2 of the PIC are logic low (0 VDC), the tristate line of the solid-state relay is connected to the GND (0 VDC). When RB0 is logic low and RB2 is logic high (+5 VDC), the tristate line is floating. When RB0 is logic high and RB1 is logic low, the tristate line is connected to the GND. When both digital outputs RB0 and RB2 are logic high, the GND signal is disconnected from NC-1 and +15 VDC is shorted from the NO-1 to NO-2 line of the solid-state relay.

The +200 VDC power supply is controlled from a digital output RB3 of the PIC through a buffer (Part No. ULN2803, STMicroelectronics, Jemstech (Pty) Ltd., Centurion, ZA, South Africa) and opto-isolator (Part No. PVI1050NS, Infineon, Jemstech (Pty) Ltd., Centurion, ZA, South Africa) circuit to switch the gate of a Metal–Oxide Semiconductor Field-Effect Transistor (MOSFET). Consequently, with the High-Voltage (HV) switched by the MOSFET, the Brick will start once the NO-2 outputs a +15 VDC signal on the tristate input of the Brick.

Appendix A.3.2. Brick Interface Board Data Acquisition

A 24 bit delta–sigma low-noise ADC (Part No. LTC2449, Analog Devices, Jemstech (Pty) Ltd., Centurion, ZA, South Africa), which provides eight differential or 16 single-ended analog input channels, is incorporated into the data acquisition of the interface boards. With reference to Figure A2b,c, the ADC has an internal 4 kHz multiplexer capable of scanning each of the eight differential channels at 500 Hz. The ADC uses the Serial Peripheral Interface (SPI) protocol for bi-directional communication with the PIC. The PIC provides a clock signal to the SCK input of the ADC to ensure data transfer synchronization. During the control of the SCK during an ADC data output cycle from the PIC, analog measurements are taken by first sending a serial bit stream request containing the resolution with the desired analog input channel to read from the Serial Data Input (SDI) of the ADC. The ADC responds by transmitting a serial bit stream from the Serial Data Output (SDO), representing the analog value measured by the particular channel. The Chip Select (CS) input of the ADC is controlled by the PIC and kept at a logic low during the analog-to-digital conversion cycle. While the conversion is in progress, the Busy line of the ADC is kept at a logic high and is fed back as a digital output to the PIC. Once the conversion is complete and the serial bit stream representing an analog value is transmitted to the PIC, a new conversion cycle is requested when a low-to-high transition is signaled on the CS input. An ADC driver that incorporates SDI, SDO, SCK, CS and Busy signals is implemented in the PIC firmware of the interface boards. The BIB data acquisition involves reading four differential inputs, namely input current (I_IN), input voltage (V_IN), output current (I_IN), output current (I_OUT) and two single-ended temperature measurements (T_MEAS_2 and T_MEAS_3) from the Brick. The serial bit stream representing an analog value is transmitted back to the Rx input of the MB in Figure A2a for monitoring and data logging in the BLA.

Appendix A.3.3. Load Interface Board Control

The LIB controls the load current of a Brick by sending digital commands received by the BLA to a Digital-to-Analog Converter (DAC) located on the DLB. As discussed in Appendix A.2, a serial signal (from either the LI1_TX or LI2_TX signals) transmitted by the BLA represents a particular task to perform by the PIC (Part No. 16F883, Microchip, Jemstech (Pty) Ltd., Centurion, ZA, South Africa). With reference to Figure A2c, the serial signal is first buffered through an opto-coupler (Part No. 6N137S, Lite-on, Jemstech (Pty) Ltd., Centurion, ZA, South Africa), which provides signal isolation from the MB and is received by an Rx input of the PIC. The Rx signal is then processed by the PIC firmware, where the DAC commands are sent over to the DLB shown in Figure A2d. The PIC digital outputs RB4, RB5, RB6 and RB7 in Figure A2c, are dedicated to controlling the Data, CLK, load and LDAC signals of the DAC (Part No. TLV5620IDR, Texas Instruments, Jemstech (Pty) Ltd., Centurion, ZA, South Africa) shown in Figure A2d. The digital conversion is

fed as the setpoint to a Voltage-Controlled Current Sink (VCCS) located on the DLB and explained in more detail in Appendix A.4.

Appendix A.3.4. Load Interface Board Data Acquisition

The LIB uses the same data acquisition hardware as explained in Appendix A.3.2. The LIB data acquisition involves reading eight differential inputs representing voltage and current measurements from their respective Bricks, where a DLB acts as resistive loads for four bricks. Similarly, a second DLB provides resistive loads for the other four Bricks. With reference to Figure A2c, the serial bit stream representing an analog value is transmitted back to the Rx input of the MB in Figure A2a for monitoring and data logging in the BLA.

*Appendix A.4. Dummy Load Board*

A Burn-in station contains two DLBs that incorporate four VCCSs, which use high-precision operational amplifiers and N-channel MOSFETs. The MOSFET is capable of a drain-to-source voltage of 200 V and supplying 50 A of drain current with a maximum on-resistance of 0.4 ohms. The amplifier acts as a comparator for error detection, which outputs a correction voltage to the gate of the MOSFET (Part No. IRFP260NPBF, Infineon, Jemstech (Pty) Ltd., Centurion, ZA, South Africa).

As illustrated in Figure A2d, a controlled voltage (O1) from the DAC is fed to the inverting input of the amplifier (Part No. OPA2197IDR, Texas Instruments, Jemstech (Pty) Ltd., Centurion, ZA, South Africa). Closed-loop feedback is established by a voltage taken from a shunt resistor (R_shunt) fed back to the non-inverting input of the amplifier. The Brick supplies a +10 VDC to the drain of the MOSFET, where the voltage over the shunt resistor (R_shunt) is proportional to the load current.

Resistors R1 and R2 in Figure A2d further provide voltage attenuation of the control signal to the inverting input of the amplifier and can be adjusted for higher current control while capacitor C1 and resistors R3 and R4 are configured as a low-pass differential RC filter to ensure loop stability. A voltage signal is fed into the gate of the MOSFET, which adjusts the on-resistance, whereby the current is sunk from the source to the drain of the MOSFET to the Ground (GND). The MOSFET dissipates heat through a cooling plate, which is water-cooled.

The thermostat (Part No. 67F110, Airpax, Jemstech (Pty) Ltd., Centurion, ZA, South Africa) switch, denoted by TS in Figure A2d, provides additional protection by closing upon a rising temperature of between 85 °C and 95 °C and consequently shorting the inverting input of the amplifier to the GND. The short to the GND leads to almost no current flowing through the drain to source of the MOSFET. Voltage divider circuits (R_shunt, R6, R7 and R9, R10, R11) are used as signal attenuators for the Brick output voltage and load current measurements fed to differential inputs of the ADC.

**Appendix B. Custom fLVPS Brick Burn-In Station Software**

*Appendix B.1. Burn-In LabVIEW Application*

With reference to Figure A3, the BLA provides control and monitoring of the Burn-in station and communicates via USB to both the MB and HV power supply using separate USB ports. The PC runs the BLA responsible for Brick identification, Brick selection, Brick control (Brick start, stop and set current load), Brick load performance (Brick input and output measurements), HV control and monitoring, Brick trip detection, Brick automatic restart, Burn-in time management and data logging.

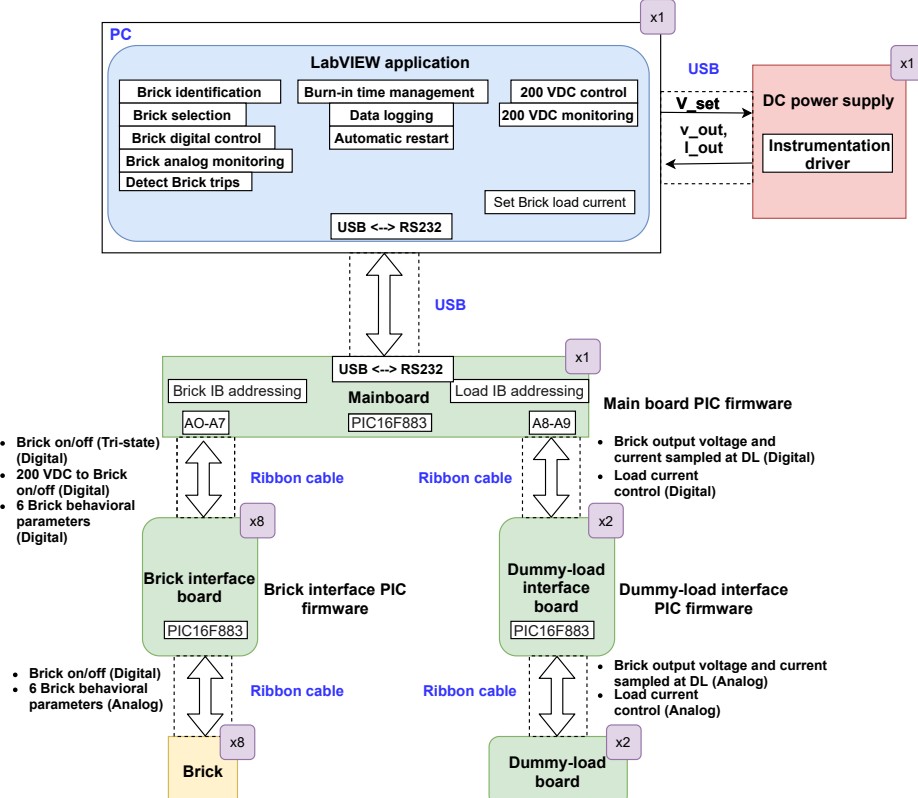

**Figure A3.** A simplified block diagram of the Burn-in station communication system. A Burn-in LabVIEW Application (BLA) running on a PC is responsible for the control, monitoring and management of the Burn-in procedure.

An addressing scheme is implemented for communication from the BLA through the MB to the interface boards, where addresses A0 to A7 represent the BIBs and addresses A8 to A9 the LIBs. The Bricks are controlled and monitored by sending serial commands to the BIB addresses A0 to A7. Similarly, Brick loads are initialised and monitored by sending serial commands to the LIB addresses A8 and A9. LIB addresses A8 and A9 control and monitor loads to Bricks 1 to 4 and Bricks 5 to 8, respectively.

The BLA mainly consists of an initialisation sequence, the Burn-in sequence and a stop sequence to end the Burn-in procedure and exit the program. The following sections present the BLA using simplified flowcharts, which describe key aspects of the BLA functional design specification. The software was originally developed by Argonne National Laboratory (ANL) in 2006 for the version 6 Bricks [12]. In conjunction with various Test-stand and Burn-in station hardware upgrade campaigns [18], the software has undergone major revisions during the ensuing years for the utilisation of the current Wits and UTA Burn-in stations.

Appendix B.1.1. Initialisation Sequence

With reference to the flowchart represented in Figure A4, two serial port communication interfaces are initialised for communication from the PC to the MB and HV power supply. The identification numbers of the IBs are checked for errors followed by a user prompt to enter the desired load current of each Brick for the Burn-in procedure. Text files are created for logging data into a file directory entered by the operator on the front panel of the BLA. Each IB is associated with an address from A0 to A9, which is polled by the BLA and controlled by the MB. Once the Brick load current commands are received from the BLA, the load currents are set by the PIC firmware of the respective LIB in Subroutine 5. Subsequently, the Bricks are sequentially started in Subroutine 6 and the HV power supply is enabled at +200 VDC. Once complete, calibration constants are initialised for

Brick measurements consisting of first-order polynomials. The time duration of the Burn-in procedure is captured by the operator before entering the main loop, as shown at the end of the flowchart. The initialise sequence is only executed once.

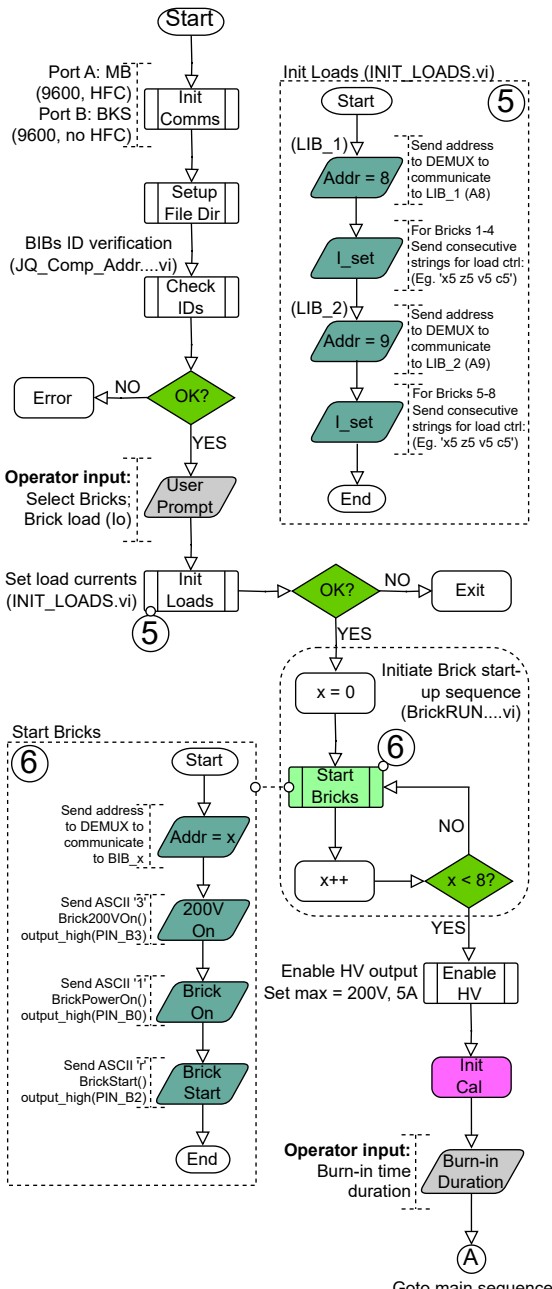

**Figure A4.** Simplified BLA initialisation sequence.

Appendix B.1.2. Main Sequence

With reference to the flowchart represented in Figure A5, the main loop starts counting down the entered Burn-in time duration. The Burn-in procedure consists of a stacked sequence structure, which cycles through each of the Brick and HV measurements until the Burn-in time has expired. A cycle is incremented once all eight Brick measurements are completed. In Subroutine 3, Brick performance measurements are read at address x, followed by a conditional statement, which checks if the Brick tripped by comparing measurements to predefined values. In Subroutine 4, the load current is set for the specified Brick and can be changed during a Burn-in procedure. In Subroutine 8, serial commands

are sent from the BLA at address x to the respective Brick to read analog values (Iin, Vin, Iout, Vout, T1, T2, T3). In Subroutine 7, if a Brick address is equal to 0–3, measurements are also taken at Address 8, representing LIB1, by sending serial commands to take analog measurements from the DLB1 (Io, Vo). Similarly, load measurements are repeated for Brick addresses 4–7. Once the measurements are read by the BLA, the values are conditioned in the software through calibration constants followed by logging measurements for analysis offline. The sequence of measuring a cycle of eight Bricks is repeated and the loop is exited once the Burn-in time has expired, followed by an entrance in the stop sequence, as shown at the end of the flowchart.

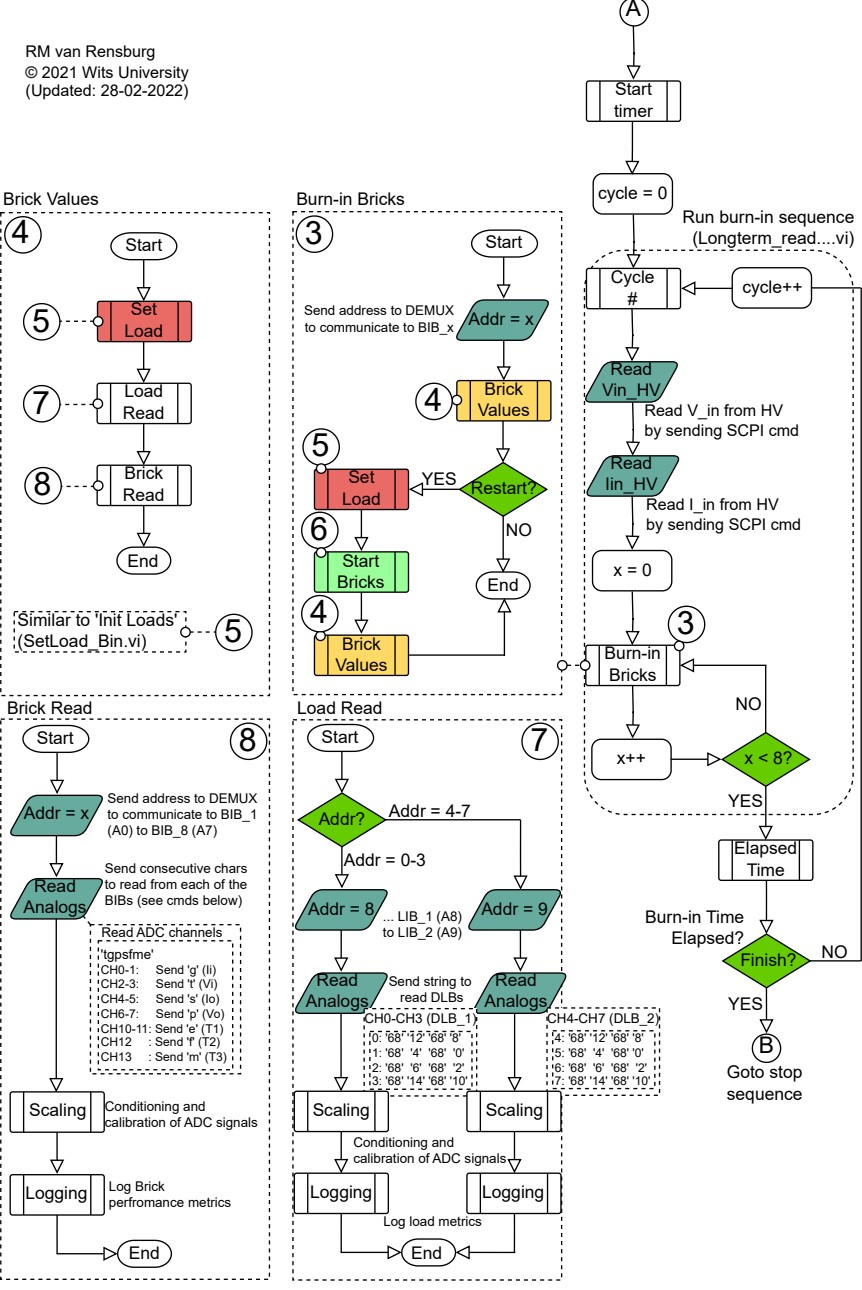

**Figure A5.** Simplified BLA main Burn-in sequence.

Appendix B.1.3. Stop Sequence

With reference to the flowchart presented in Figure A6, the stop sequence disables the HV power supply and stops each of the eight Bricks one at a time. The serial ports

are closed and reports are generated for data analysis offline. The stop sequence is only executed once.

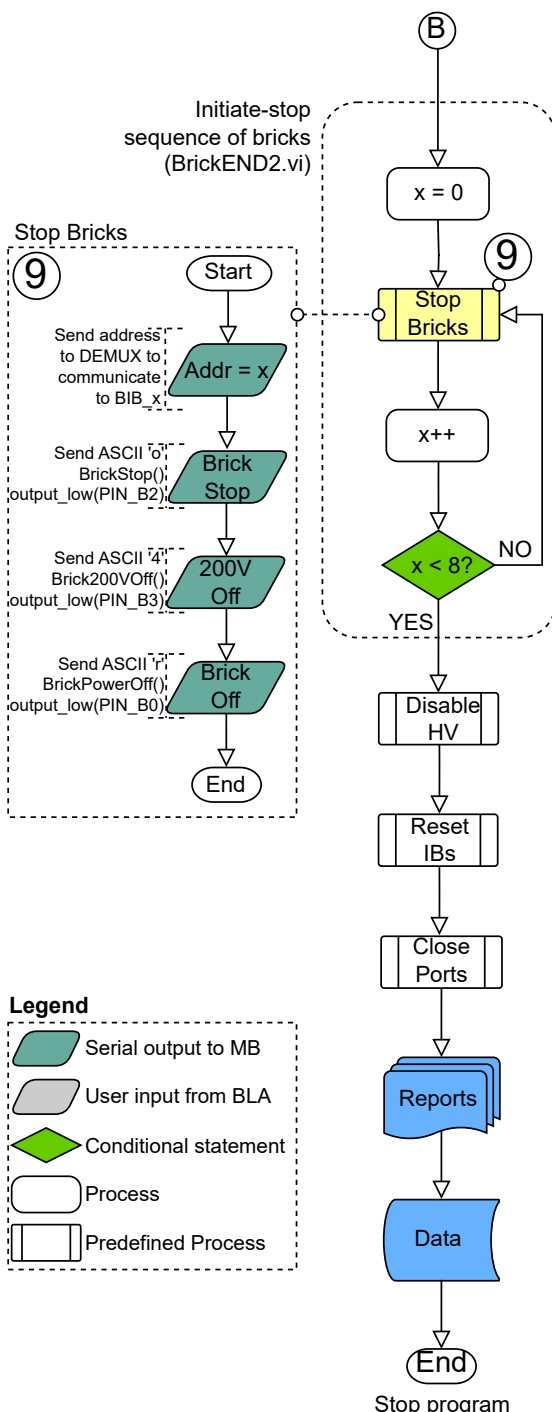

**Figure A6.** Simplified BLA stop sequence.

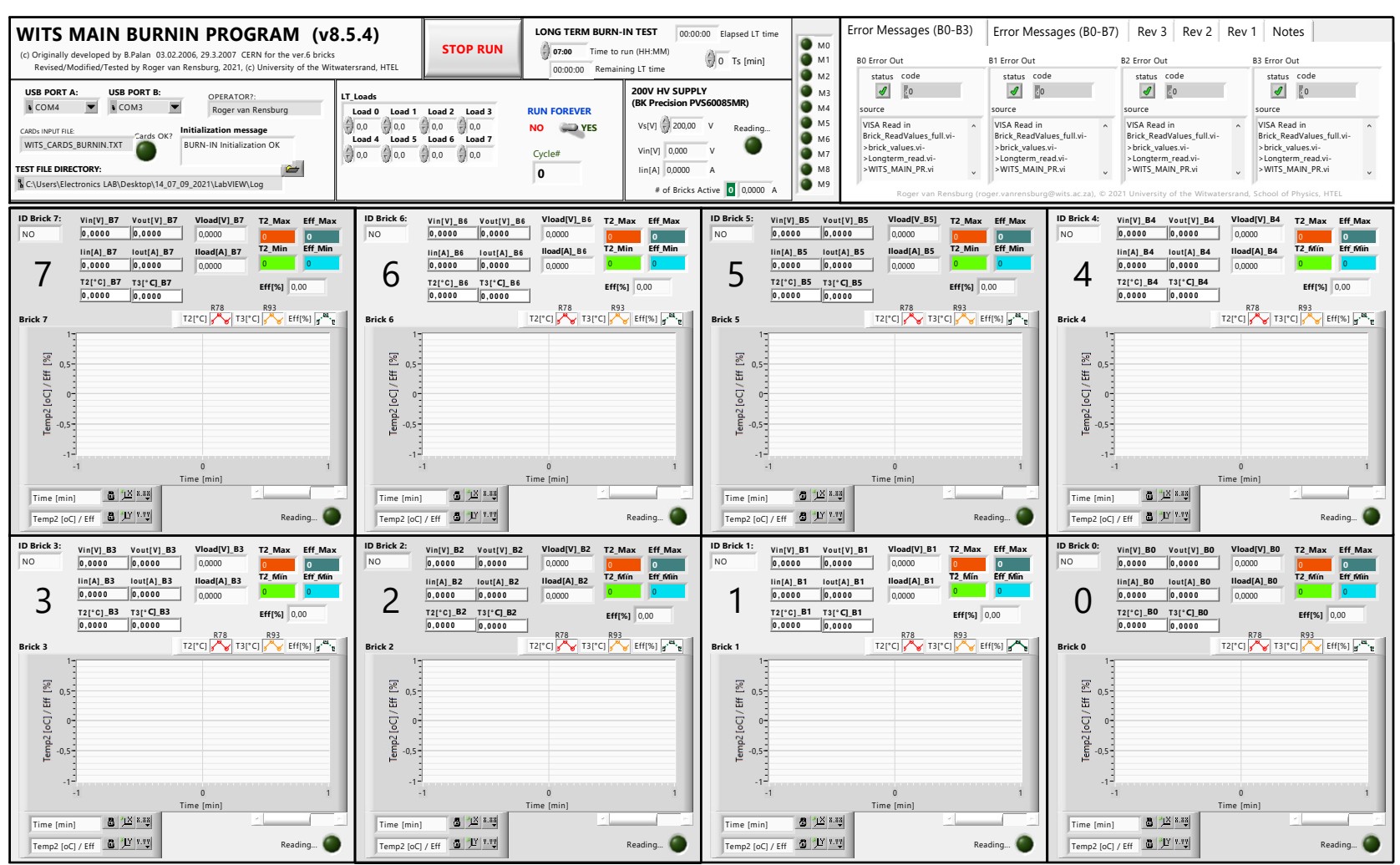

**Figure A7.** Wits Graphical User Interface (GUI) of the Burn-in LabVIEW Application (BLA) illustrating the control and monitoring of the Brick Burn-in procedure. Both Wits and UTA institutions make use of a different GUI (BLA Front Panel), but the underlying code (BLA Block Diagram) functionality is identical.

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
