# Peer review of "A Burn-In Apparatus for the ATLAS Tile Calorimeter Phase-II Upgrade Transformer-Coupled Buck Converters"

_instruments, doi:10.3390/instruments7040041_

Round 1

Reviewer 1 Report

Comments and Suggestions for Authors

This is a nice paper that provides a wealth of information on Burn-In apparatus for ATLAS. The authors were very thorough in spelling out the technical details of various considerations and the effort that went into designing this apparatus along with successful demonstration.

The paper reads too technical to me and at times it is easy to get lost in acronyms. I feel more elaboration or broader context is missing at certain places. I have made suggestions at these instances for improvements. The conclusions can benefit from a more quantitative and qualitative set of remarks, even plots, from the tests performed so far if there are enough/reasonable statistics. Some more elaboration on pre-production and production responsibilities, and how that is shared b/n various institutes e.g. UTA, would be useful.

L2: Abstract: define the acronym “LHC”
L20: provide a reference for HL-LHC if available
L20: define CERN
L21: space between 4000 and fb^-1 (always give space between number and unit; you do this in some places so be consistent)
L25: space b/n “m” and “<“
L48-50: is there a reference you can cite for V6.5.4, their issues and performance? it would be informational to include it if there is a public reference.
L169: define PIC
L183: I assume UM232R is a model number of the converter? It seems you have indicated model numbers at other places too e.g. 194, 205, 221, 239 etc. It is a bit odd to provide model numbers without a reference or vendor name. It is perhaps to map the description in the text with what is in the figures. Some clarification somewhere about this would be nice and/or consider including citations to key equipment if model numbers are referenced.
L187: define UART
L396: I would suggest briefly elaborating on v7.5.0 here, their issues, their performance and how that motivated improvements to the new version.
L414: There needs to be some discussion on how the 8 hour window was chosen and how that is the right duration for the test. I suggest adding a couple of sentence to motivate this.
Section 9: This needs some improvement and I would strongly recommend adding some qualitative and quantitative results on the tests performed thus far with the burn in station. It could be along the lines of we have tested these versions of bricks (v8? other versions?) and these many, and these are the results of the tests from these batches and provide qualitative description of key observations and understanding from those tests. This would be a great addition to this section. If there are any plots that can be made from the tests that shows how well e.g. the over temperature  or other parameters were sustained, that would be also be valuable.
L492: it is mentioned that UT Arlington will also become a burn-in site. Is it just a stand-by site or will share QA responsibilities with Wits? What is the capacity of the burn-in station at Wits? Provide some more context to this since you bring this up.

Comments on the Quality of English Language

Overall, the paper reads well and is well done language-wise. I only have minor comments below:

L21: “n order to” —> “in order to”
Figure 1: caption: “experiment” —> “experiment’s”
L55: such —> such as
L97: on “the” detector
L193: feds —> feeds
General comment: try to avoid using acronyms in captions; figures should be able to stand on their own without someone having to dig into the text to understand what each acronym means. It is probably impossible to get rid of all but I would try to go with full expansions in captions where possible.
L352: “to” repeated twice
L401: “the” repeated twice
L405: “in” thermal equilibrium
L451: remove “type”

Author Response

Dear Reviewer,

All authors would like to thank you for your detailed review comments which have been greatly received.

All comments have been addressed in the latest draft of the paper which has now taken a very different form to that which was originally submitted.

Best

Reviewer 2 Report

Comments and Suggestions for Authors

I congratulate the authors on a tremendous and important effort in developing hardware for the TileCal upgrade.

My general comments are that the paper, while a tremendous amount of work is not particularly easy to read. I would suggest that some of the details be moved to appendices so that the main outcome of the burn-in process can be clearly gleaned by a reader. One way that I could envision this happening is by moving some of the details, e.g., sections 5-7 to appendices, and briefly summarizing the results in the body of the text.

Another area for improvement is the introduction. While the HL-LHC is described, there are missing references and motivation. References listed could be better distributed (e.g., ref 1-5 could be spread out to describe the LH-LHC more directly). I think there are TileCal references that are missing, e.g., https://arxiv.org/abs/1007.5423

The description of TileCal physics needs references. For instance, you discuss the opportunity to use the calorimeter for tau lepton reconstruction, jet reconstruction, L1 triggers. There are papers on most of these topics that should be cited. I would also like to see some discussion of how muons deposit energy in TileCal (though it can be very brief). One good reference is: http://cdsweb.cern.ch/record/1196071

section 2: It may be nice to show a picture of the bricks here, if you can. Without knowing what it looks like, it's hard to assess if that request makes sense. If you have one it would be nice.

Figure 12: there is a figure that seems (I think) to not be discussed in the text. It's an interesting figure, and if you show it, it should be discussed. The figure also merits some labels for what is being shown.

section 8.2: were you able to correlate the temperature measurements from the thermistor with the IR camera in figure 12? that may be worth mentioning.

Section 9: is there a target for the fraction of bricks that need to pass burn in? That would be interesting to note, particularly for the purposes of contingency in production.

Comments on the Quality of English Language

L21: typo, should say "in order to"

L50: I suppose trips should be expanded, as this is jargon

Figure 13 caption: typo "texorpdfstring"

L466: you give a reference to a thermistor, but it's just a number. Suggest explaining the manufacturer etc.

References 8 and 11 are not complete

Author Response

Dear Reviewer,

All authors would like to thank you for your detailed review comments which have been greatly received.

All comments have been addressed in the latest draft of the paper which has now taken a very different form to that which was originally submitted.

With regards to the  request for some coverage of the TileCal physics we have added a citation to the TileCal run1 performance paper which presents the topic in greater detail which would fall out of the scope of this paper.

Best

Round 2

Reviewer 1 Report

Comments and Suggestions for Authors

The updated version looks very good and the authors have addressed all my comments with great thoroughness. I don't have any new comments and I am happy to sign off on this.

Reviewer 2 Report

Comments and Suggestions for Authors

I did a detailed reading, and suggested some re-organization. In the next draft I did a cursory reading and the re-organization dramatically improves the reading. The responses also look reasonable.

Comments on the Quality of English Language

none